# Transcriptome Sequencing and Metabolome Analysis Reveals the Molecular Mechanism of Drought Stress in Millet

**DOI:** 10.3390/ijms231810792

**Published:** 2022-09-16

**Authors:** Xiaoning Cao, Yulu Hu, Jian Song, Hui Feng, Junjie Wang, Ling Chen, Lun Wang, Xianmin Diao, Yan Wan, Sichen Liu, Zhijun Qiao

**Affiliations:** 1Center for Agricultural Genetic Resources Research, Shanxi Agricultural University, Taiyuan 030031, China; 2Key Laboratory of Crop Gene Resources and Germplasm Enhancement on Loess Plateau, Ministry of Agriculture, Taiyuan 030031, China; 3Institute of Crop Sciences, Chinese Academy of Agricultural Sciences, Beijing 100081, China; 4Key Laboratory of Coarse Cereal Processing, Ministry of Agriculture and Rural Affairs, Chengdu University, Chengdu 610106, China

**Keywords:** *Panicum miliaceum* L., drought stress, transcriptome, metabolome, transcription factors, plant hormone

## Abstract

As one of the oldest agricultural crops in China, millet (*Panicum miliaceum*) has powerful drought tolerance. In this study, transcriptome and metabolome analyses of ‘Hequ Red millet’ (HQ) and ‘Yanshu No.10’ (YS10) millet after 6 h of drought stress were performed. Transcriptome characteristics of drought stress in HQ and YS10 were characterized by Pacbio full-length transcriptome sequencing. The pathway analysis of the differentially expressed genes (DEGs) showed that the highly enriched categories were related to starch and sucrose metabolism, pyruvate metabolism, metabolic pathways, and the biosynthesis of secondary metabolites when the two millet varieties were subjected to drought stress. Under drought stress, 245 genes related to energy metabolism were found to show significant changes between the two strains. Further analysis showed that 219 genes related to plant hormone signal transduction also participated in the drought response. In addition, numerous genes involved in anthocyanin metabolism and photosynthesis were confirmed to be related to drought stress, and these genes showed significant differential expression and played an important role in anthocyanin metabolism and photosynthesis. Moreover, we identified 496 transcription factors related to drought stress, which came from 10 different transcription factor families, such as bHLH, C3H, MYB, and WRKY. Further analysis showed that many key genes related to energy metabolism, such as citrate synthase, isocitrate dehydrogenase, and ATP synthase, showed significant upregulation, and most of the structural genes involved in anthocyanin biosynthesis also showed significant upregulation in both strains. Most genes related to plant hormone signal transduction showed upregulated expression, while many JA and SA signaling pathway-related genes were downregulated. Metabolome analysis was performed on ‘Hequ red millet’ (HQ) and ‘Yanshu 10’ (YS10), a total of 2082 differential metabolites (DEMs) were identified. These findings indicate that energy metabolism, anthocyanins, photosynthesis, and plant hormones are closely related to the drought resistance of millet and adapt to adversity by precisely regulating the levels of various molecular pathways.

## 1. Introduction

Millet (*Panicum miliaceum* L.) is an important economic crop of the Gramineae family, used as a food crop with a long history of cultivation that originated in China. It is a highly nutritious cereal grain used for human consumption, bird seed, and/or ethanol production [1,2]. It has a short growth period, drought resistance and barren resistance [3]. Millet is one of the most important multigrain crops in arid and semi-arid areas of northern China, and is of great significance in dryland ecological agriculture, diversity of grain production and food security [4]. Drought is one of the most frequent and severe abiotic stress factors, and its disadvantages affect plant growth and crop productivity in many arid and semiarid regions [5]. As the grain with the highest water use efficiency [6], broomcorn millet has strong drought resistance and is an ideal system to study the mechanism of drought tolerance and reverse resistance. Predecessors have done a lot of research on the osmotic regulation [7], physiological response [8] and changes of rhizosphere microorganisms [9] of broomcorn millet in response to drought stress. Most millet planting areas are concentrated in arid or semiarid areas, and water is the main factor that affects millet yield. Improving the water use efficiency of crops via genetic means is the focus of current crop breeding research, and this requires us to systematically clarify the molecular mechanism of millet response to drought stress.

Despite the complexity of drought tolerance, tremendous progress has been made in understanding the drought-adaptive mechanisms of plants [10,11]. Plant drought tolerance occurs mainly through changes in plant morphology and growth, involving complex gene expression regulation and molecules in the adaptation process pathway. Drought stress reduces photosynthesis by cutting down the unit leaf area and photosynthetic rate (PR) [12,13]. The reduced PR occurs mainly through stomatal closure or metabolic impairment. Continued photosynthetic light reactions during drought stress under limited intercellular CO2 concentrations result in the accumulation of reduced photosynthetic electron transport components, which can potentially reduce molecular oxygen, leading to the production of reactive oxygen species (ROS) [14,15,16].

Additionally, major phytohormones, such as abscisic acid (ABA), cytokinin (CK), gibberellic acid (GA), auxin, and ethylene, regulate diverse processes that enable plant adaptation to drought stress [17,18]. Studies have shown that ABA can control the closure of plant stomatal, reduce plant growth, and allow plants to adapt to drought stress. CK delays the senescence and death of plant leaves, thereby allowing plants to adapt to drought stress [19]. Auxin might participate in the positive regulation of drought stress resistance through regulation of root architecture, ABA-responsive genes expression, ROS metabolism, and metabolic homeostasis [20]. Exogenous IAA improved drought tolerance of white clover possibly due to endogenous plant hormone concentration changes (ABA and JA) and modulation of genes involving in drought stress response and leaf senescence [21]. Exogenous application of GA3 improved the water stress tolerance in maize plants by maintaining membrane permeability, enhancing chlorophyll concentration and leaf relative water content (LRWC) [22]. Arabidopsis plants overexpressing AtERF019 showed increased tolerance to water deficiency that could be explained by a lower transpiration rate [23]. The involvement of MeJA improved the drought tolerance of soybean by modulating the membrane lipid peroxidation and antioxidant activities [24]. Plant hormones do not work in isolation, but instead interact and regulate each other’s biosynthesis and response [25].

In a short time, drought will change the physiological metabolism of plants to adapt to adversity, while long-term drought will affect the morphological structure and biomass distribution of each plant organ [26,27]. When water becomes the limiting factor affecting plant growth, in order to ensure the maintenance of life, plants will sacrifice their growth rate and produce a large number of secondary metabolites to protect their growth under adverse conditions. Anthocyanins are an important type of secondary metabolites in plants [28]. In addition to attracting insects to spread pollen, they are also widely involved in the process of UV resistance and drought resistance. In Arabidopsis and cherries, drought causes the accumulation of anthocyanins, thereby increasing the plant’s resistance to drought [29,30,31]. Transcription factors are the regulatory factors in plant response to abiotic and biotic stresses. It plays an important role in plant drought response [32]. The literature shows that the WRKY family of transcription factors also play an important role in plant drought stress [33,34]. GhWRKY1-like may act as a positive regulator in Arabidopsis tolerance to drought via directly interacting with the promoters of AtNCED2, AtNCED5, AtNCED6, and AtNCED9, to promote ABA biosynthesis [35]. Drought stress induces the accumulation of OsbHLH130, which in turn activates the expression of OsWIH2, which improves drought tolerance in rice by participating in epidermal wax biosynthesis and reducing water loss rate and ROS accumulation [36]. NAC17 transcription factor contributes to drought tolerance by modulating lignin accumulation in rice [37]. The transcription factor ZmMYB-CC10 enhances tolerance to drought stress by directly activating ZmAPX4 expression in maize, thereby reducing H_2_O_2_ content [38].

In this study, transcriptome sequencing and quantification of non-targeted metabolites were performed in millet strains with different drought resistance (HQ and YS10), and the molecular pathways and gene expression profiles related to millet drought resistance were determined. Through analysis, many DEGs were individually identified between the two strains, and these DEGs combined with the differential metabolites (DEMs) in the subsequent analysis were further analyzed. These results provide an opportunity to elucidate the molecular mechanism underlying the drought resistance of millet.

## 2. Results

### 2.1. Transcriptome Characteristics of HQ and YS10 under Drought Stress

The collection of full-length transcriptome data is based on the third-generation sequencing platform of PacBio sequence. By filtering original data, we deleted connector and original offline data less than 50 bp in length to obtain 37,997,585 subreads based on 46.88 G clean reads. The average length of subreads is 1231 bp. After further analysis, 908,964 full-length non-chimeric reads (FLNC) were obtained, of which the average length of FLNC was 2163 bp. The third-generation sequencing technology represented by PacBio has the advantage of long read length, but the single-base error rate of this technology is very high. To reduce the high error rate, Illumina data was used to make corrections. After correcting the data, a total of 514,570 transcriptomes were identified. After further alignment of the reads to the genome, 112,437 and 87,330 Isoforms (Figure A1A) were identified in HQ and YS10 respectively. In order to obtain comprehensive gene annotations, we used multiple databases to annotate Isoforms. The 100,362 and 78,008 isoforms were annotated in HQ and YS10 respectively, and the overall annotation rate was 89.32%. According to the mapping result of the transcript and the reference genome, the reads aligned to the unannotated region of the reference genome gtf file are defined as new genes. With benefit from high-quality data generated by full-length transcriptome sequencing, we conducted the identification of lncRNA. Based on transcript length and gene coding ability, more than 10,000 new lncRNAs were identified in HQ and YS10, most of which belong to LincRNA and sense_overlapping lncRNA (Table 1).

### 2.2. Differential Gene Identification and Functional Analysis under Drought Stress

The differences in drought resistance between the HQ and YS10 strains can cause differences in transcriptome dynamics. DEGs during the progression of stress showed stress responsiveness and their putative roles during drought tolerance. Through paired comparison, a total of 36,121 DEGs in millet were identified, of which 18,146, and 17,975 DEGs, were found between HQ and YS10 strains with drought stress, respectively. A total of 8578 genes were upregulated and 9568 genes were downregulated under the drought stress treatment in HQ (Figure A2B). To further explore the biological functions of the DEGs, the DEGs were evaluated via KEGG pathway analysis. The top 20 pathways for the most prominent differentially expressed genes are listed (Figure A2C). Interestingly, we found that most of the DEGs identified in two strains were enriched in the same molecular pathways, including the biosynthesis of secondary metabolites, starch and sucrose metabolism, pyruvate metabolism, and metabolic pathways. The above analysis results show that the two millet lines, HQ and YS10, have great similarities in response to drought stress.

### 2.3. The Effects of Drought Stress on Energy Metabolism

By analyzing the changes in genes related to sugar metabolism, significant changes in carbohydrate metabolism pathways were observed. We found that the metabolic pathways related to starch and sucrose were activated, while glycolysis and TCA cycle activities were significantly reduced. A total of 126 genes involved in sucrose and starch metabolism were identified as having significant differential expression in the HD and YD groups, including phosphomannomutase, glucose-6-phosphate isomerase 1, beta-glucosidase, zingiberene synthase, UDP-glycosyltransferase, sucrose synthase, haloacid dehalogenase, starch synthase, and beta-amylase (Figure 1). In general, with the exception of beta-glucosidase, the genes showed upregulated expression for the HD and YD comparisons. This shows that under unfavorable conditions, plants attempt to use more carbohydrates to deal with these unfavorable environments. The upregulated expression of UDP-glycosyltransferase may be closely related to plant secondary metabolism, which confers stronger drought resistance to millet. Furthermore, 127 genes involved in glycolysis and the TCA cycle were also identified, including pyruvate kinase, isocitrate dehydrogenase, succinate dehydrogenase, malate dehydrogenase, pyruvate carboxylase, hexokinase, aldolase, phosphofructokinase 3, pyruvate dehydrogenase E1 alpha, ATP citrate lyase, pyruvate dehydrogenase decarboxylase, citrate synthase, and aldehyde dehydrogenase. Among them, the key enzymes in the TCA cycle, 4 citrate synthases and 8 isocitrate dehydrogenases, were all upregulated in the HD and YD, and the expression of these genes in the YD group was higher than that in the HD group. In addition, hexokinase (HK), the rate-limiting enzyme of glycolysis, also showed significant differential expression. The 16 and 15 HK genes were significantly downregulated in the HD and YD, respectively. In addition, most malate dehydrogenase (MDH) and succinate dehydrogenase (SDH) genes were downregulated in the HD and YD, and only a small portion of MDH and SDH genes showed upregulated expression. Interestingly, the expression levels of 3 upregulated MDH and 4 SDH genes in YD were significantly higher than their expression levels in the HD group.

### 2.4. Dynamic Changes in the Transcriptome of Plant Hormone Signal Transduction

The genes involved in auxin transport, local biosynthesis and signaling pathways may modify metabolism to establish an adaptation mechanism in plants subjected to drought stress. It has been reported that ABA, JA, cytokinins, and GA, promote light-induced anthocyanin biosynthesis, and anthocyanin can enhance the ability of plants to resist drought [39,40]. In our study, 220 genes related to 7 plant hormone (PH) signaling pathways were determined to be differentially expressed in the HQ and YS10 strains (Figure 2). Among them, 25 DEGs were related to the ABA signaling pathway, 54 DEGs were related to the auxin signaling pathway, 78 DEGs were related to the ethylene signaling pathway, 27 DEGs were related to the cytokinin signaling pathway, 5 DEGs were related to the BR signaling pathway, 21 DEGs were related to the GA signaling pathway, and 10 DEGs were related to the JA and SA signaling pathways. In general, most of the PH-related genes were upregulated in the HD and YD. The difference is that among the auxin signaling pathway-related genes, 61.11% of these genes (33/54) were upregulated in the HD, while only 46.29% of these genes (25/54) were upregulated in the YD. Among the genes related to the ethylene signaling pathway, 69% (54/78) were upregulated in the HD, while the ratio in the YD was 60% (47/78). Although most PH-related genes were upregulated, 90% (9/10) of the genes related to the JA and SA signaling pathways were significantly downregulated in the HD and YD group. In summary, ABA, AUX/IAA, ethylene, GA, BR, cytokinin, and other hormone signaling pathways, are enhanced under drought stress, while the JA and SA signaling pathways are inhibited, which may be directly related to the drought resistance of millet.

### 2.5. Increased Anthocyanin Metabolism Improves the Drought Resistance of HQ and YS10

Studies have shown that anthocyanin content is significantly related to the enhancement of plant drought resistance. Therefore, the regulation of genes related to anthocyanin biosynthesis may lead to changes in the content of anthocyanins in broomcorn millet, thereby affecting its drought resistance. In this study, we identified 97 structural genes associated with anthocyanin biosynthesis (Figure 3). These genes include PAL: phenylalanine ammonia lyase; C4H: cinnamate 4-hydroxylase; 4CL: 4-coumaroyl: CoA ligase; CHS: chalcone synthase; CHI: chalcone isomerase; F3H: flavanone 3-hydroxylase; DFR: dihydroflavonol-4-reductase; OMT: O-methyltransferase; and GST: glutathione S-transferase. Among them, 4 PAL, 9 4CL, 2 CHS, 8 F3H, 1 DFR, 1 OMT, 2 CHI, and 25 GST genes, were significantly upregulated in the HD and YD, while 1 4CL, 3 F3H, 2 OMT, and 8 GST genes, were upregulated in only the HD group. These data indicate that both the HQ and YS10 strains significantly enhanced the metabolic activity of anthocyanin biosynthesis when subjected to drought stress. Moreover, the expression of structural genes associated with anthocyanin biosynthesis in the HQ strain was significantly higher than that in the YD strain, indicating that the regulation of these genes was significantly related to drought resistance in millet.

### 2.6. Expression Profiling of Photosynthesis-Related Genes

Photosystem I is the second photosystem in the photosynthetic light reactions in algae, plants, and some bacteria. Photosystem I is an integral membrane protein complex that uses light energy to produce the high energy carriers ATP and NADPH. PSI comprises more than 110 cofactors, which is significantly more than those involved in photosystem II. Our data show that the changes in the genes related to the photosystem may also be related to drought stress in millet. Thirty-five genes related to the photosystem and photoreaction were identified, including 13 photosystem II reaction center subunits (PSB), 9 photosystem I reaction center subunits (PSA), 4 LHC genes, and 9 ATP synthase genes (Figure 4). Among them, 9 PSB genes were upregulated in the HD group compared to the HC group, and 12 were upregulated in the YD group compared to the YC group. Moreover, almost all PSA and LHC genes were downregulated in the HD vs. HC comparison, and 7 upregulated genes and 6 downregulated genes were identified in the YD. ATP synthase participates in oxidative phosphorylation and photosynthetic phosphorylation and synthesizes ATP under the impetus of transmembrane proton kinetic potential. Our data show that there are 5 ATPS genes that are jointly upregulated in the HD and YD, and there are also 4 ATPS genes that are downregulated. Further analysis found that 5 upregulated ATPSs had higher expression levels in the YD group than in the HD group, and 4 downregulated ATPSs had lower expression levels in the HD group than in the YD group. This result indicates that the differential expression levels of PSA and ATPS genes are one of the important reasons for the differences in drought resistance in the HQ and YS10 lines.

### 2.7. Identification and Expression Analysis of Transcription Factors Related to Drought Stress

Transcription factors play an important regulatory role in plant drought tolerance. In this study, a total of 496 transcription factors were identified as hypothetical regulators of drought resistance in millet (Figure 5). These transcription factors include bHLHs (basic helix-loop-helix), bZIPs (basic region-leucine zipper), C2H2s (C2H2 zinc-finger proteins), C3Hs (Cys3His zinc finger proteins), FAR1s (far-red impaired response 1), WD40s, MYBs (MYB domain proteins), NACs (NAM/ATAF/CUC), TCPs (TCP proteins), and WRKYs (WRKY proteins). We found that 86.23% (94/109) of WRKY family transcription factors were upregulated in the HD vs. HC comparison, and only 59.63% (65/109) of WRKY family transcription factors were upregulated in the YD. Similarly, 84.61% (55/65) of NAC family transcription factors were upregulated in the HD vs. HC comparison. In contrast, only 63.07% (41/65) of NAC family transcription factors were upregulated in the YD. Among the other transcription factor families, the HD and YD groups showed almost the same expression pattern. In total, 58.55% (65/111), 61.21%, 66.66%, 42.21%, 62.34%, 100%, 57.42%, and 67.59% of MYB, bHLH, WD40, TCP, bZIP, C2H2, C3H, and FAR1 family TFs, showed significant upregulation, respectively, in both the HD and YD. Overall, more than 70% of the 496 TF genes were significantly upregulated in the HD and YD groups, and the upregulation of these transcription factors initiated the activation of downstream molecular pathways, thereby enhancing drought resistance. In addition, WRKY and NAC family TFs are clearly key transcription factors for the difference in drought resistance between the HD and YD groups. These TFs show a very large difference in the number of activations induced by drought, which gives them different gene regulatory networks.

### 2.8. Overview of the Metabolomes of the HQ and YS10 Strains

Based on high-resolution mass spectrometry (HRMS) detection technology, the non-targeted metabolome can detect as many molecular characteristic peaks in the sample as possible. A combination of the high-quality mzCloud database and Chemspider database can identify metabolites in biological systems to the greatest extent. In this study, based on HRMS technology, non-targeted quantitative analysis of metabolites in 20 samples was performed. A total of 4831 metabolites were identified, including 2364 metabolites in negative ion mode and 2467 metabolites in positive ion mode. The PCA of metabolite content showed that these samples showed good repeatability (Figure 6A,B). Furthermore, these metabolites were annotated based on the HMDB database, and the results showed that only 10% of the metabolites were annotated. Most of these metabolites are lipids and lipid-like molecules (Figure 6C).

In order to further determine the metabolites that may play a key role in drought stress, a difference analysis based on pairwise comparisons was carried out. A total of 2082 differential metabolites (DEMs) were identified, including 928 negative ion mode differential metabolites and 1154 positive ion mode differential metabolites (Figure A2A). A total of 871 DEMs were identified in the HD vs. HC comparison, and 994 DEMs were found in the YD vs. YC comparison. The global expression pattern of these DEMs is presented in Figure A2B. Unlike the similarity of the transcriptome expression pattern, the metabolites in the HQ and YS10 strains showed specific expression patterns. Further functional analysis showed that the DEMs in the HD vs. HC comparison were closely related to energy metabolism-related pathways, such as the citric acid cycle (TCA cycle), oxidative phosphorylation, glyoxylate and dicarboxylate metabolism and pyruvate metabolism (Figure A2C), while the DEMs in the YD were related to secondary metabolism and polysaccharide metabolism, including the biosynthesis of secondary metabolites, starch and sucrose metabolism and indole alkaloid biosynthesis (Figure A2D).

In addition, we performed an association analysis on the transcriptome and metabolome data. Through integrated analysis of the association analysis results, the metabolic pathways shared by the DEGs and DEMs were identified. Similarly, the DEGs/DEMs in the HD were mostly related to energy metabolism (Figure A3A), while the DEGs/DEMs in the YD were more likely to be related to secondary metabolism (Figure A3B).

### 2.9. qRT-PCR Verification of the RNA-Seq Data

To validate the RNA-seq data, 16 indispensable genes related to key metabolic pathways were chosen for expression validation by qRT-PCR analysis (Figure 7). These genes are involved in anthocyanin biosynthesis, plant hormone signal transduction, and energy metabolism. The qRT-PCR analysis results showed that the expression of these genes displayed patterns that were very similar to those of the FPKM values from sequencing under the corresponding treatments. These results indicated that the RNA-seq data were reliable.

## 3. Discussion

When growing under drought conditions, plants try to adjust their adaptive mechanisms to maintain their cell state through osmotic adjustment and the absorption of water, and increase protoplasmic resistance so that they can escape, avoid, or tolerate drought stress [41]. In this study, through the joint analysis of the transcriptome and metabolome of two millet varieties with different drought resistances, HQ and YS10, the molecular mechanism of millet drought tolerance (DR) was initially clarified. Moreover, by comparing the differences in the genes related to the key metabolic pathways between the two lines, the regulatory factors that led to their differences in drought resistance were discovered. Our results show that the drought resistance of millet is mainly related to energy metabolism, photosynthesis, anthocyanin metabolism, plant hormone signal transduction, and DR-related transcription factors. Under drought stress, the activity of anthocyanin metabolism is significantly enhanced, and most of the genes related to plant hormone signal transduction, such as ABA, CK, and GA, also show upregulated expression. In contrast to other crops, we found that the expression of PSII-related genes and key enzymes of the TCA cycle did not decrease under drought stress but increased their expression to a certain extent.

Generally, drought stress reduced the net photosynthesis rate and supply capacity of the plant source. Studies have shown that drought stress in maize etc. will generally show a sharp decline in the expression of photosynthesis-related genes, thereby reducing the production of ROS in plants and reducing damage [42]. Interestingly, we found that all Psb protein genes in the PSII core complex were upregulated in both YS10 and HQ. These genes include PSBW, PSB27-1, PSB28, PSBT, etc., among which PSB28 is the constituent protein of the PS II reaction center and directly participates in the photosynthetic reaction [43]. PSB27-1 is involved in the repair of photodamaged photosystem II (PSII), while PSBW plays an important role in stabilizing the PSII dimer [44,45]. Our data showed PSB27-1 upregulation in both the HD vs. HC and YD vs. YC comparisons. PSB27-1 had 1.57-, 2.83-, and 1.41-fold changes in the HD vs. HC, YD vs. YC, and HD vs. YD comparisons, respectively. Unlike YS10, the HQ strain maintained a higher expression of PSB27-1 under normal growth conditions. However, under drought stress, the expression of the PSB27-1 gene in the YS10 strain increased significantly. Long-term drought conditions can cause damage to the photosystem of plants, and millet improves the stability of its photosystem by increasing the photosystem repair protein. At the same time, PSB27-1 gene expression in the HQ strain was higher than that in the YS10 strain, which may be related to its stronger DR. There were PSBT and PSBW photosystem proteins that showed a change trend similar to that of PS27-1 [46,47,48]. These genes also play a role in the formation and stability of the PSII system. Although the expression levels of these genes in YS10 are low under normal growth conditions, after being induced by drought, the expression levels of these genes in the HQ strain will rapidly increase to similar levels. Unlike PSII, the situation in PSI is different. PSA2, PSA3 [49,50], and other key factors that promote the stable assembly of PSI were significantly downregulated in the HD vs. HC and YD vs. YC comparisons. Moreover, PSAN, PSAH, and PSAO, were significantly upregulated in the YS10 strain, but there was no change in the HQ strain. PSAN may function in mediating the binding of the antenna complexes to the PSI reaction center and core antenna [51,52], while PSAH is responsible for the docking of the LHC I antenna complex and the PSI core complex [53]. In summary, the factors that promote the stability of the photosystem in the PSII and PSI systems are upregulated under drought stress to cope with possible photosystem damage. Additionally, we found that ATPG, ATPD, and other genes encoding ATP synthase subunits, were significantly upregulated in HQ and YS10. ATP synthase (ATPS) produces ATP from ADP in the presence of a proton gradient on the membrane. ATPS is essential for photosynthesis, probably by facilitating electron transport in both photosystems I and II. The upregulated expression of ATPG and ATPD in millet provides sufficient energy support under drought stress and enhances drought resistance [54]. Interestingly, the expression of ATPG/D in the YS10 strain was higher than that in HQ, which may indicate that YS10 needs more energy to maintain its normal development and growth. In addition to photosynthesis, sugar metabolism is also a core pathway of energy metabolism in plants under drought stress.

We found that beta-amylase (BAM) genes were upregulated in HQ and YS10, while starch synthase (SS) was downregulated [55,56,57]. Millet copes with energy deficiency under water shortage conditions by strengthening starch catabolism and inhibiting anabolism. Moreover, sucrose synthase 1 (SUS1), a sucrose-cleaving enzyme that provides UDP-glucose and fructose for various metabolic pathways, showed drastic downregulation in both HQ and YS10 [40,58]. This result also shows that millet acts similar to other plants by separating sucrose, and the pathway provides intermediates and energy sources. In short, HQ and YS10 provide an energy source by enhancing the catabolism of starch and sucrose and inhibiting its anabolic and other necessary physiological activities. Glycolysis is a prerequisite reaction for the aerobic metabolism of glucose, and 90% (9/10) of the rate-limiting enzyme hexokinase (HK) was significantly upregulated in HQ and YS10 [59,60]. In addition, phosphofructokinase 3 (PFK3) is one of the keys to glycolysis, and this enzyme also showed significant upregulation in HQ and YS10 [61]. Among the key genes in the TCA cycle, IDH and MDH showed downregulated expression, while CS showed upregulated expression. Overall, glycolysis was enhanced, and the TCA cycle was inhibited. This may indicate that millet may perform more anaerobic respiration under drought stress, while aerobic respiration is inhibited.

Environmental factors regulate the color reaction of anthocyanins by inducing the expression of genes related to the anthocyanin synthesis pathway in plants. Low temperatures can induce the accumulation of anthocyanins and increase their contents [62,63]; high temperatures will accelerate the degradation of anthocyanins [31,64,65]. Studies have shown that a lack of water can promote the accumulation of anthocyanins [39,66]. Expression analysis found that the lack of water increased the expression of the anthocyanin synthesis pathway genes F3H, DFR, UFGT, and GST. Our data show that PAL, 4CL, CHS, F3H, DFR, OMT, CHI, GST, and other anthocyanin biosynthetic structural genes are expressed in HQ and YS10, and their expression in HQ was higher than that in YS10. This result shows that anthocyanins play an important role in millet drought tolerance and provides an explanation for the difference in drought tolerance between HQ and YS10. Drought stress can cause changes in the contents of a variety of endogenous plant hormones. Although their mechanisms of action are different, the changes in various plant hormones are not isolated but instead interact and influence each other to coordinate the regulation of plant growth and development under adversity. It is generally believed that ABA, CK, JA/SA, and other hormones, are enhancers of plant drought tolerance [67,68,69,70]. In this study, we found that the ABA receptors PYL5, PYL9, and PYL10, were all upregulated in both HQ and YS10. Auxin has been thought to play an important role in primordium establishment and growth [20,71]. In our data, IAA30, IAA21, IAA17, IAA31, IAA21, and IAA13, were all downregulated, while IAA1 and IAA6 were upregulated. Studies have shown that overexpression of IAA13 slows the growth of lateral roots and floral organs [72]. The downregulated expression of IAA13 may promote the development of millet transverse roots, which may enhance drought tolerance. The regulation of plant growth and development by different hormones creates a very delicate and complex network, and the synergistic effects of hormones is an important regulation method for drought tolerance. In addition, transcription factors play a key regulatory role in the plant response to drought stress. Studies have shown that the transcription factors involved in drought stress in plants mainly include NAC transcription factors, bZIP (basic-domain leucine-zipper) transcription factors, zinc finger protein transcription factors, MYB transcription factors, WRKY transcription factors, and TIFY transcription factors [73,74,75,76,77]. Through the identification of transcription factors, we identified 496 transcription factors of 10 types in millet. These transcription factors are considered putative regulators of drought tolerance. Studies have shown that members of the WRKY transcription factor family, such as WRKY16/WRKY16-A, WRKY17, WRKY17, WRKY19-C, WRKY24, WRKY59, WRKY61, and WRKY82, are positive regulators of wheat drought tolerance. We identified more than 70% of the 496 TFs that were significantly upregulated in HD and YD, and WRKTs and the NAC transcription factor superfamily were considered to be the most important transcription factor families in HQ and YS10 drought tolerance. Over 86.62% and 84.43% of the WRKY and NAC transcription factor superfamily members were significantly upregulated in HQ, while the proportions in YS10 were 59.63% and 63.07%, respectively. This shows that the WRKY and NAC family transcription factors may also be two of the main factors leading to the difference in drought tolerance between HQ and YS10.

In the combined analysis of the transcriptome and metabolome, the DEGs/DEMs in the HD vs. HC comparison were mostly related to energy metabolism, while the DEGs/DEMs in the YD vs. YC comparison were more related to secondary metabolism. In general, HQ and YS10 have great similarities in drought tolerance mechanisms, but their metabolic differences in transcription factors and anthocyanin synthesis lead to differences in drought tolerance.

## 4. Materials and Methods

### 4.1. Plant Materials, Cultivation and Treatment

The drought-resistant variety Hequ red millet and the drought-sensitive variety Yanshu No. 10, which have been confirmed in previous studies [78], were subjected to drought stress treatment. Hequ red millet is a landrace variety widely grown in arid and semi-arid areas in northwestern Shanxi, and Yanshu 10 is an excellent millet line selected after hybridization. Plant the two strains in a plants hydroponics growth box containing nutrient solution. Then, place the box in an incubator with 16 h sunshine duration, 25 °C/22 °C (day/night) ambient temperature and 60% relative humidity. After 18 days, the millet was transferred to 15% PEG-6000 solution for stress treatment. After 6 h, the leaves of the millet were collected and stored in liquid nitrogen and used for transcriptome and metabolome detection. The samples included the Hequ red millet control group (HC) and drought stress treatment group (HD), the Yanshu No. 10 control group (YC) and stress treatment group (YD). Each group was repeated three times.

### 4.2. Transcriptome Sequencing and Metabolome Detection

There were 3 biological replicates for each group of transcriptome sequencing, and 5 biological replicates for each group of metabolome sequencing. The extraction of total RNA and metabolites was performed according to the manufacturer’s instructions. After obtaining mRNAs and the SMRT library was prepared, there was sequencing by PacBio platform. At the same time, a library was constructed with sequenced on the Illumina Nova-Seq 6000 platform. The metabolites were injected into the LC-MS/MS system for analysis. LC-MS/MS analyses were performed using a Vanquish UHPLC system (Thermo Fisher, Waltham, MA, USA) in Novogene Co., Ltd. (Beijing, China)coupled with an Orbitrap Q Exactive series mass spectrometer (Thermo Fisher).

### 4.3. Bioinformatics Analysis

Full-length transcriptome sequencing data were processed using the SMRTlink 5.0 software. Circular consensus sequence (CCS) was generated from subread BAM files. Reference genome and gene model annotation files were downloaded from the genome website directly. The index of the reference genome was built using Hisat2 (v2.1.0), and paired-end clean reads were aligned to the reference genome using Hisat2. HTSeq v0.6.1 was used to count the read numbers mapped to each gene [79]. Then, the FPKM of each gene was calculated based on the length of the gene and read count mapped to this gene. Differential expression analysis of two groups (with three biological replicates per condition) was performed using the DESeq2 (1.18.0) [80]. Genes with an adjusted *p*-value < 0.05 and fold changes > 2 were considered as DEGs. The metabolites were annotated using the KEGG database (http://www.genome.jp/kegg/, accessed on 6 May 2020), and HMDB database (http://www.hmdb.ca/, accessed on 10 May 2020). Principal component analysis (PCA) and partial least squares discriminant analysis (PLS-DA) were performed with the metaX softwareX2.0 [81,82]. The metabolites with VIP > 1, *p*-value < 0.05 and fold change ≥ 2 were considered to be differential metabolites. Volcano plots, cluster heat map and correlation analysis (method = Pearson) were performed by R 3.6.1. We used the KOBAS software 3.0 to test the statistical enrichment of differentially expressed genes and differential metabolites in KEGG pathways. The KEGG pathways with corrected *p*-values of less than 0.05 were considered significantly enriched by differentially expressed genes.

### 4.4. qRT-PCR

The TaKaRa MiniBEST Plant RNA Extraction Kit was used to extract RNA from all samples according to the manufacturer’s instructions. A NanoDrop 20000 (Thermo Scientific, Pittsburgh, PA, USA) was used to detect the RNA concentration and purity, and high-quality RNA was selected for reverse transcription. The PrimeScript™ RT reagent Kit with gDNA Eraser was used for reverse transcription of RNA, and the synthesized first-strand cDNA was used as a template for qRT-PCR. qRT-PCR was performed on a CFX96 Real-Time PCR Detection System (Bio-Rad, Hercules, CA, USA). Each sample was repeated three times, and three technical replicates were performed.

## 5. Conclusions

These findings indicate that energy metabolism, anthocyanins, photosynthesis, and plant hormones, are closely related to the drought resistance of millet and adapt to adversity by precisely regulating the levels of various molecular pathways.

## Figures and Tables

**Figure 1 ijms-23-10792-f001:**
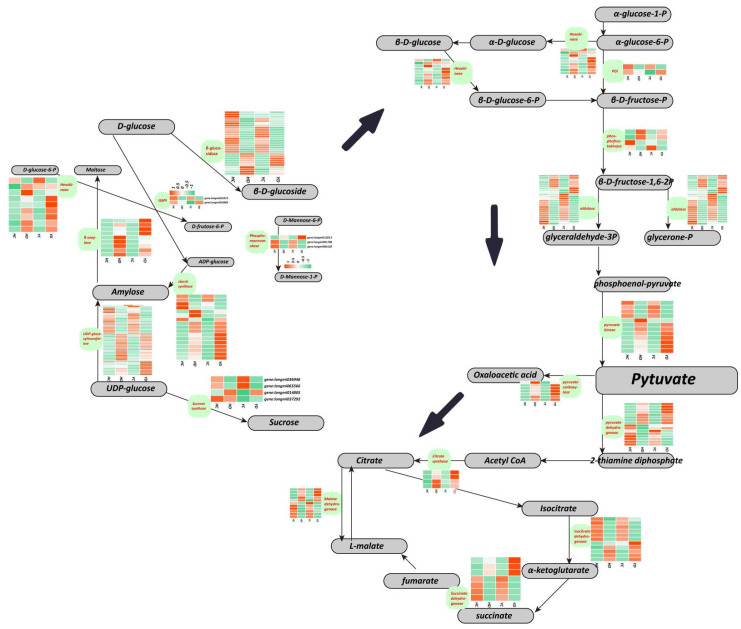
The expression profile of genes involved in sugar metabolism. These genes include starch and sucrose metabolism, glycolysis, and the TCA cycle, involving 21 related enzymes and regulatory factors.

**Figure 2 ijms-23-10792-f002:**
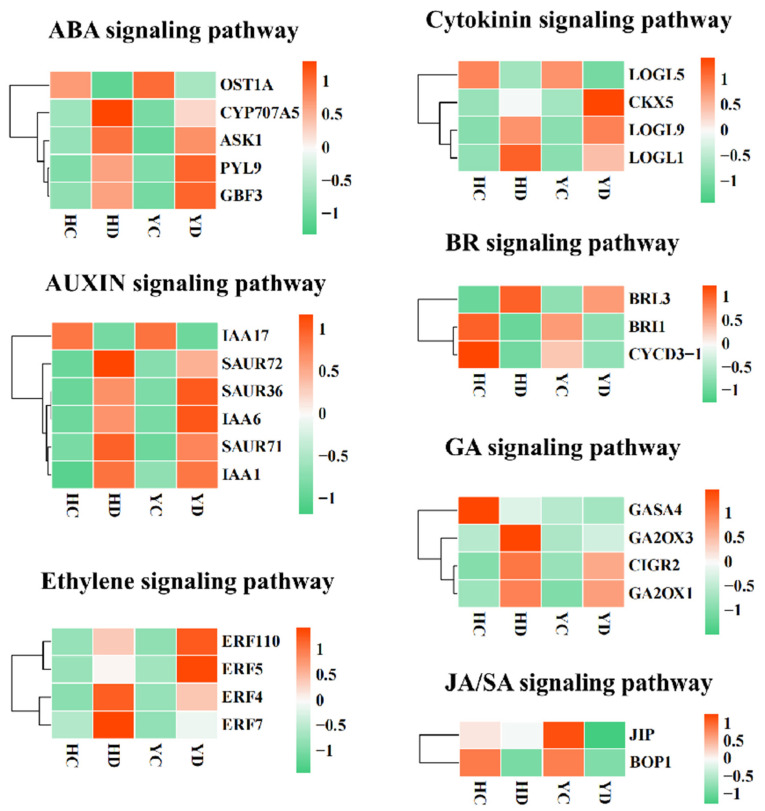
The expression profiles of the genes related to plant hormone signal transduction in HQ and YS10. These hormone pathways include the ABA signaling pathway, auxin signaling pathway, ethylene signaling pathway, cytokinin signaling pathway, BR signaling pathway, GA signaling pathway, and JA and SA signaling pathways.

**Figure 3 ijms-23-10792-f003:**
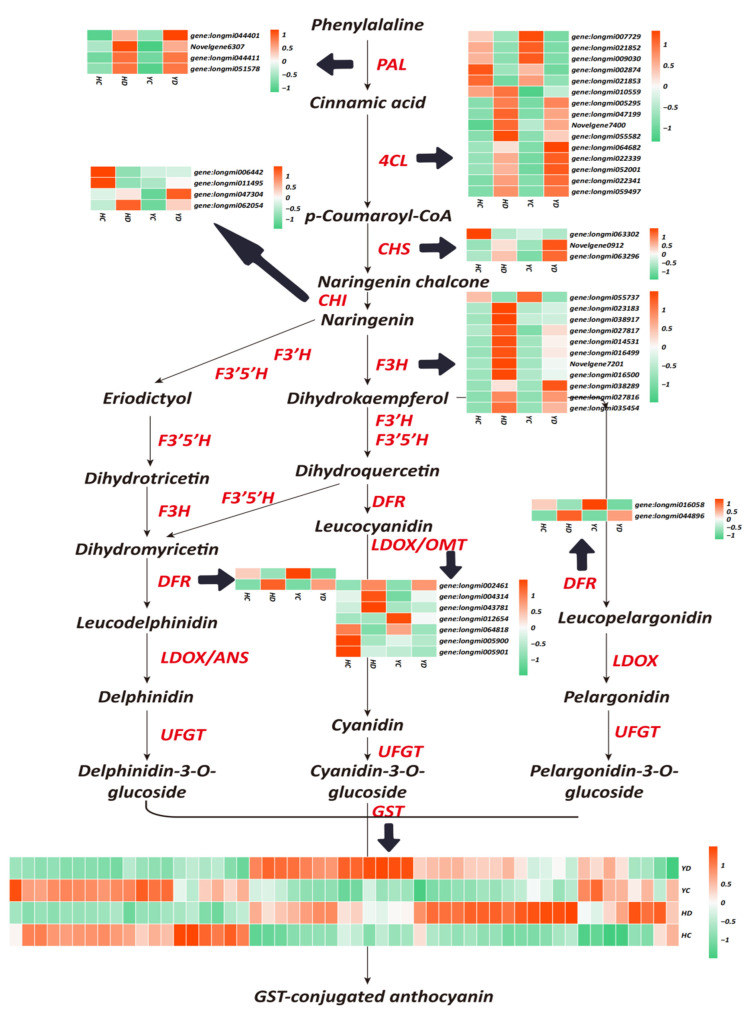
Heat map diagrams of the relative expression levels of anthocyanin biosynthesis-related structural genes in response to DR. PAL: phenylalanine ammonia lyase; C4H: cinnamate 4-hydroxylase; 4CL: 4-coumaroyl-CoA ligase; CHS: chalcone synthase; CHI: chalcone isomerase; F3H: flavanone 3-hydroxylase; DFR: dihydroflavonol-4-reductase; OMT: O-methyltransferase; and GST: glutathione S-transferase.

**Figure 4 ijms-23-10792-f004:**
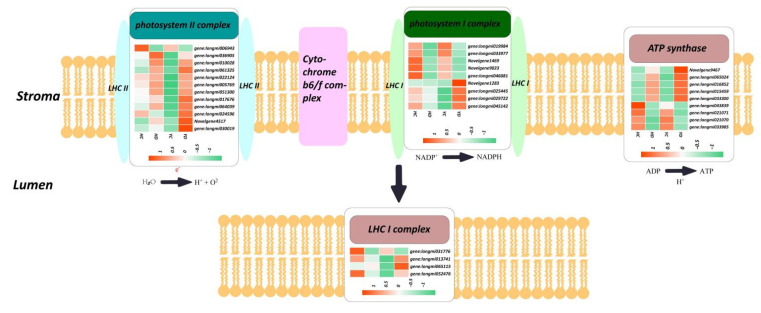
The expression profile of photosystem-related genes, including the PSA, PSB, LHC, ATPG, and ATPD genes.

**Figure 5 ijms-23-10792-f005:**
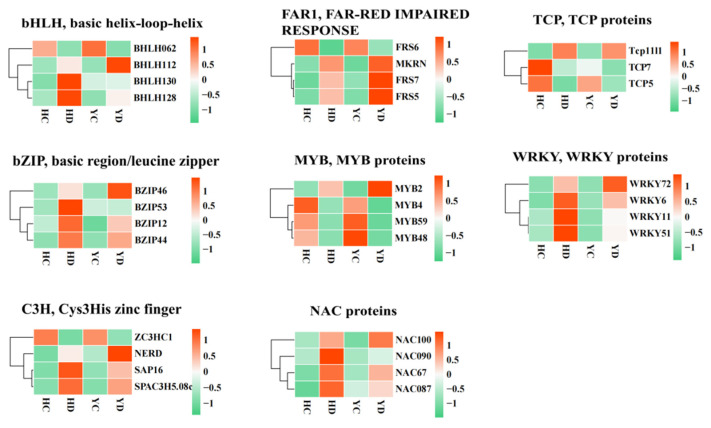
Heat map diagrams of the relative expression levels of TFs annotated in anthocyanin biosynthesis in millet: bHLH, basic helix-loop-helix; bZIP, basic region/leucine zipper; C2H2, C2H2 zinc-finger proteins; C3H, Cys3His zinc finger; FAR1, far-red impaired response 1; TCP, TCP proteins; WD40, WD40 repeat proteins; MYB, MYB proteins; NAC, NAC proteins; and WRKY, WRKY proteins.

**Figure 6 ijms-23-10792-f006:**
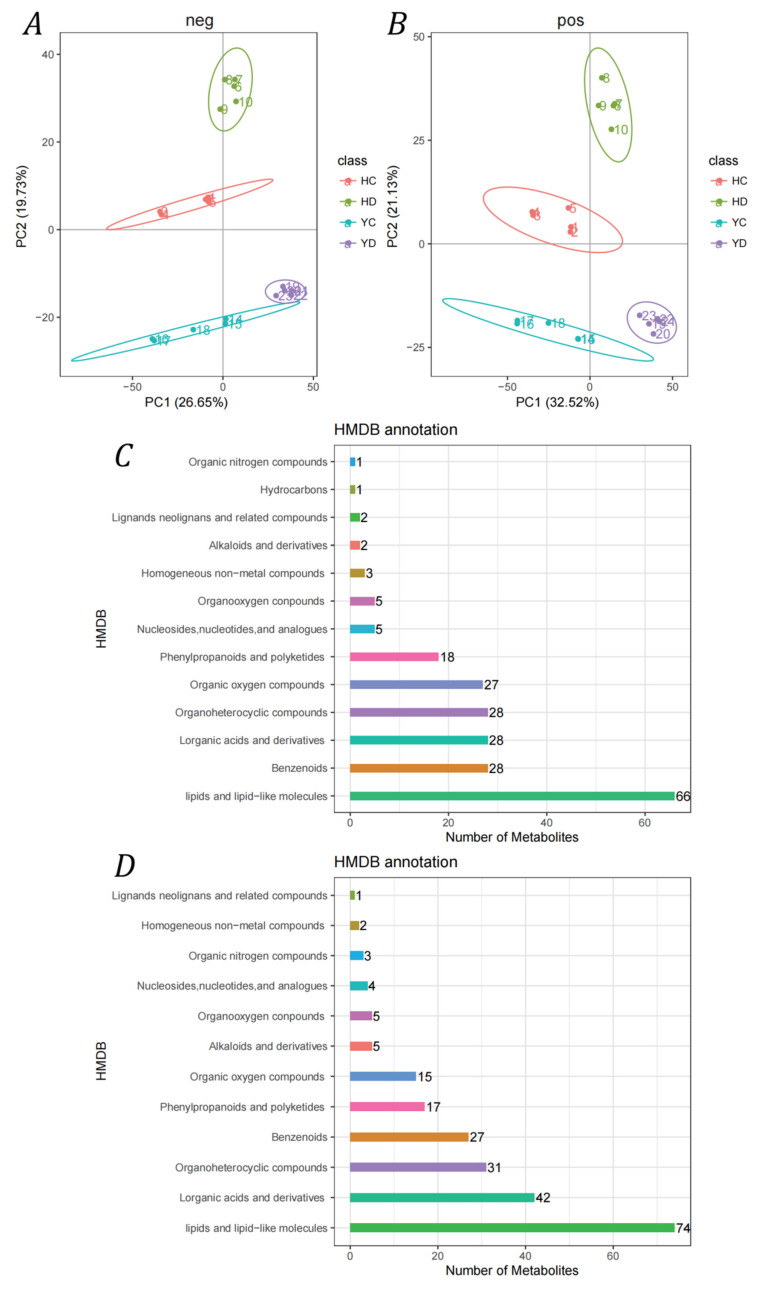
PCA of quantitative metabolite data and HMDB annotation of metabolites. (**A**,**B**): PCA of metabolites in positive ion mode and negative ion mode. (**C**,**D**): HMDB database annotations for metabolites in positive ion mode and negative ion mode.

**Figure 7 ijms-23-10792-f007:**
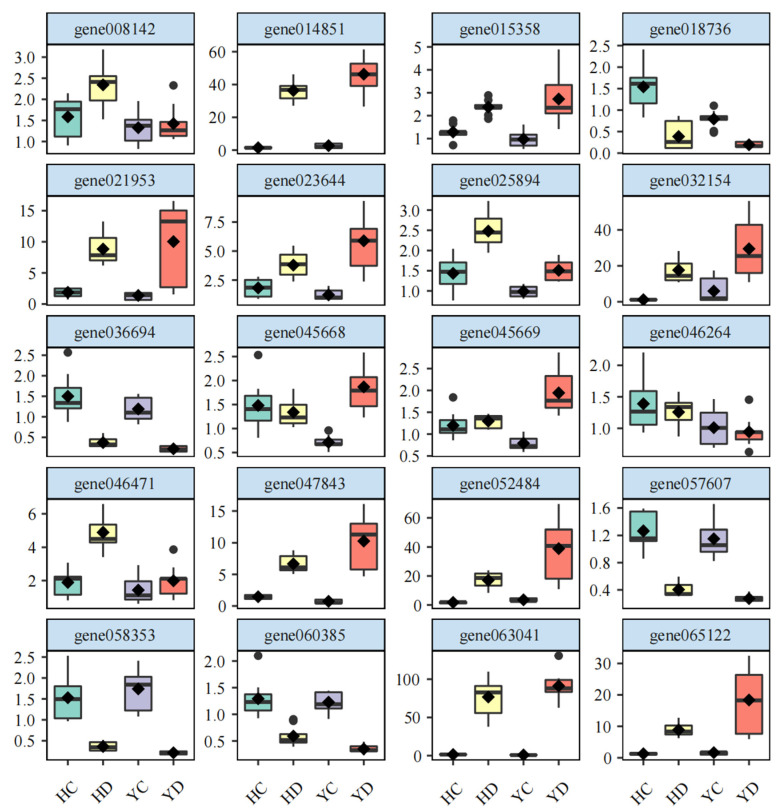
qRT-PCR verification of 16 genes related to key metabolic pathways.

**Table 1 ijms-23-10792-t001:** The overview of full-length transcriptome.

	HQ	YS
Subreads		
Subreads base(G)	26.11	20.77
Subreads number	20,615,482	17,382,103
Average subreads length	1267	1196
FLNC		
FLNC_number	531,759	377,205
Mean_length	2152	2174
Total number after correct	288,698	225,872
Isoforms number	112,437	87,330
Annotation num		
NR	4721	3656
SwissProt	2910	2269
KEGG	4446	3422
KOG	2081	1605
GO	3208	2545
NT	5153	4038
PFAM	3208	2545
lncRNA		
Sense_overlapping	3342	430
Sense_intronic	241	2868
LincRNA	3701	2939
Antisense	538	172

## Data Availability

The raw data have been uploaded to the SRA database with the number SRP282714.

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
