# Peer review of "Transcriptome Sequencing and Metabolome Analysis Reveals the Molecular Mechanism of Drought Stress in Millet"

_ijms, 2022, doi:10.3390/ijms231810792_

Round 1
Reviewer 1 Report
The manuscript by Cao et al. presents to reveal the molecular mechanism of drought stress in millet by transcriptome and metabolome analysis. It certainly offered some knowledge about how millet responding to drought stress, here are some comments:
1.In the first paragraph the author mentions“It has a short growth period (60–100 days) and is widely distributed around the world due to its drought resistance.”,but there is no mention of the drought resistance and drought tolerance of millet, or why millet was chosen as the object of drought resistance research, and some related research progress of millet's drought resistance was not mentioned. Please add it.
2.Line 111,(fig. A1A) ,This expression looks unreasonable, can it be changed to "figure A1-A".
3.Why did you choose these 16 genes for expression validation by qRT-PCR analysis?
4.In 4.1.Plant materials, cultivation and treatment,the author just proposed two different cultivars of millet with different drought resistance, but did not explain the biological characteristics of the two cultivars clearly.
5.In References,The format of references should be uniform.
Author Response
Response to Reviewer 1 Comments
Point1:In the first paragraph the author mentions“It has a short growth period (60–100 days) and is widely distributed around the world due to its drought resistance.”,but there is no mention of the drought resistance and drought tolerance of millet, or why millet was chosen as the object of drought resistance research, and some related research progress of millet's drought resistance was not mentioned. Please add it.
Response 1:The comments made by the reviewers are great suggestions, and I have added them to the text:
It has a short growth period, drought resistance and barren resistance[3].Millet is one of the most important multigrain crops in arid and semi-arid areas of northern China, and is of great significance in dryland ecological agriculture, diversity of grain production and food security[4].Drought is one of the most frequent and severe abiotic stress factors, and its disadvantages affect plant growth and crop productivity in many arid and semiarid regions[5]. As the grain with the highest water use efficiency [6], broomcorn millet has strong drought resistance and is an ideal system to study the mechanism of drought tolerance and reverse resistance. Predecessors have done a lot of research on the osmotic regulation [7], physiological response [8] and changes of rhizosphere microorganisms [9] of broomcorn millet in response to drought stress.Most millet planting areas are concentrated in arid or semiarid areas, and water is the main factor that affects millet yield. Improving the water use efficiency of crops via genetic means is the focus of current crop breeding research, and this requires us to systematically clarify the molecular mechanism of millet response to drought stress.
Point2: Line 111,(fig. A1A) ,This expression looks unreasonable, can it be changed to "figure A1-A".
Response 2:The comments from the reviewers are very useful, I have made changes in the text
Point 3: Why did you choose these 16 genes for expression validation by qRT-PCR analysis?
Response 3: The comments from the reviewers are very valuable:
The 16 genes selected for validation are all indispensable genes for key metabolic pathways in this study.
Point 4:In 4.1.Plant materials, cultivation and treatment,the author just proposed two different cultivars of millet with different drought resistance, but did not explain the biological characteristics of the two cultivars clearly.
Response 4: The comments made by the reviewers are very necessary, and I have added them in the text:
Hequ red millet is a landrace variety widely grown in arid and semi-arid areas in northwestern Shanxi, and Yanshu 10 is an excellent millet line selected after hybridization.
Point 5:In References,The format of references should be uniform.
Response 5:It is really a great suggestion as Reviewer pointed out that all references in the article have been edited.

Reviewer 2 Report
Overall authors have done good job to described millet response at transcriptomic and metabolome level under drought conditions. Authors have analyzed two millet lines by RNA sequencing and quantification of non-target metabolome analysis under drought conditions. They have used ‘Hequ Red millet’ (HQ and 'Yanshu No.10’ (YS10) millet lines for transcriptome and metabolome analysis to identify pathways which are involved in drought stress. They have shown that when two millets were grown under drought conditions, differentially expressed genes (DEGs) involved in pyruvate metabolism, starch and sucrose metabolism, metabolic pathways, and the biosynthesis of secondary metabolites. Authors have shown 245 genes relevant to energy metabolism and 219 genes related to plant hormone signal transduction in response to drought. They also shown that several genes involved in anthocyanin metabolism and photosynthesis. They also identified 496 transcription factors belong to families such as bHLH, C3H, MYB, and WRKY under drought conditions among two lines. However, there are few miner concerns mentioned below:
Abstract: in my opinion, although abstract is well written, but they emphasis on DEGs involved in metabolites, I would say authors should add 1-2 sentences metabolome comparative analysis.
Introduction: it could be improved by adding information of plant hormones and transcription factors involved in drought resistance.
Results:
Figure caption should be clarified and font size in figures should big enough to visualize. Please revise font size in all figures.
Figure 6. please follow above suggestion. Fig. 6c and D is not readable.
Figure 7. please revise figure title and add figure caption in detail. Figure caption should be independent to describe results in figure.
Also revise figure caption in Figure A2.
Please carefully set your references

Author Response
Point 1:Abstract: in my opinion, although abstract is well written, but they emphasis on DEGs involved in metabolites, I would say authors should add 1-2 sentences metabolome comparative analysis.
Response1: The comments made by the reviewers are very necessary, and I have added them to the text:
Metabolome analysis was performed on 'Hequ red millet' (HQ) and 'Yanshu 10' (YS10),a total of 2082 differential metabolites (DEMs) were identified.
Point 2: Introduction: it could be improved by adding information of plant hormones and transcription factors involved in drought resistance.
Response2:The comments from the reviewers are very useful,and I have added them to the text:
Auxin might participate in the positive regulation of drought stress resistance,through regulation of root architecture, ABA-responsive genes expression, ROS metabolism, and metabolic homeostasis[20].Exogenous IAA improved drought tolerance of white clover possibly due to endogenous plant hormone concentration changes(ABA and JA) and modulation of genes involving in drought stress response and leaf senescence[21]. Exogenous application of GA3 improved the water stress tolerance in maize plants by maintaining membrane permeability, enhancing chlorophyll concentrationand leaf relative water content(LRWC) [22]. Arabidopsis plants overexpressing AtERF019 showed increased tolerance to water deficiency that could be explained by a lower transpiration rate[23].The involvement of MeJA improved the drought tolerance of soybean by modulating the membrane lipid peroxidation and antioxidant activities[24].Plant hormones do not work in isolation, but instead interact and regulate each other's biosynthesis and response[25].
The literature shows that the WRKY family of transcription factors also play an important role in plant drought stress[33,34].GhWRKY1-like may act as a positive regulator in Arabidopsis tolerance to drought via directly interacting with the promoters of AtNCED2, AtNCED5, AtNCED6 and AtNCED9 to promote ABA biosynthesis[35].Drought stress induces the accumulation of OsbHLH130, which in turn activates the expression of OsWIH2, which improves drought tolerance in rice by participating in epidermal wax biosynthesis and reducing water loss rate and ROS accumulation[36].NAC17 transcription factor contributes to drought tolerance by modulating lignin accumulation in rice[37].The transcription factor ZmMYB-CC10 enhances tolerance to drought stress by directly activating ZmAPX4 expression in maize, thereby reducing H2O2 content[38].
Point 3: Results: Figure caption should be clarified and font size in figures should big enough to visualize. Please revise font size in all figures.
Response3:The comments made by the reviewers are very necessary, and I have revised all the figures in the text.
Point 4: Figure 6. please follow above suggestion. Fig. 6c and D is not readable.
Response4:The comments made by the reviewers are very necessary, and I have revised the figures in the text.
Point 5:Figure 7. please revise figure title and add figure caption in detail. Figure caption should be independent to describe results in figure.
Response5: The comments made by the reviewers are very necessary, and I have revised all the figures in the text.
Point 6:Also revise figure caption in Figure A2.
Response6:The comments made by the reviewers are very necessary, and I have revised all the figures in the text.
Point 7: Please carefully set your references
Response7:It is really a great suggestion as reviewer pointed out that all references in the article have been edited.
